# Research data management in agricultural sciences in Germany: We are not yet where we want to be

**Matthias Senft**[1]*, **Ulrike Stahl**[2], **Nikolai Svoboda**[3]

**1** Leibniz Institute for Agricultural Engineering and Bioeconomy (ATB), Potsdam, Germany, **2** Julius Kühn Institute (JKI)—Federal Research Centre for Cultivated Plants, Quedlinburg, Germany, **3** Leibniz Centre for Agricultural Landscape Research (ZALF), Müncheberg, Germany

* matthias.senft@julius-kuehn.de

**Data Availability Statement:** The questionnaire and data set with the survey results are published under a CC-BY license in the repository OpenAgrar (doi: 10.5073/20211013-105447).

## Abstract

To meet the future challenges and foster integrated and holistic research approaches in agricultural sciences, new and sustainable methods in research data management (RDM) are needed. The involvement of scientific users is a critical success factor for their development. We conducted an online survey in 2020 among different user groups in agricultural sciences about their RDM practices and needs. In total, the questionnaire contained 52 questions on information about produced and (re-)used data, data quality aspects, information about the use of standards, publication practices and legal aspects of agricultural research data, the current situation in RDM in regards to awareness, consulting and curricula as well as needs of the agricultural community in respect to future developments. We received 196 (partially) completed questionnaires from data providers, data users, infrastructure and information service providers. In addition to the diversity in the research data landscape of agricultural sciences in Germany, the study reveals challenges, deficits and uncertainties in handling research data in agricultural sciences standing in the way of access and efficient reuse of valuable research data. However, the study also suggests and discusses potential solutions to enhance data publications, facilitate and secure data re-use, ensure data quality and develop services (i.e. training, support and bundling services). Therefore, our research article provides the basis for the development of common RDM, future infrastructures and services needed to foster the cultural change in handling research data across agricultural sciences in Germany and beyond.

## Introduction

Agricultural science, more than other domains, operates at the nexus of science and business facing far reaching challenges such as an increasing demand for high quality agricultural products impacting the environment [1]. In addition to the loss of biodiversity and degradation of natural resources, agriculture is one of the major drivers of climate change (24% of total greenhouse gas (GHG) emissions [2]) through activities such as the application of urea and

**Funding:** The authors received no specific funding for this work.

**Competing interests:** The authors have declared that no competing interests exist.

agricultural lime, enteric fermentation, rice cultivation, use of fertilizers, and degradation of soils with high natural carbon content [3–6]. However, state of the art research data management and clever combination of various useful data can, among other things, help to move from a GHG source to a carbon sink. Such complex relationships cannot be addressed by isolated research, but require well-connected scientific sub-disciplines that also go beyond agricultural research. To understand the carbon balance of a landscape, data from long-term soil monitoring, farm management, climate and the environment must be analyzed in a model-based way. As a result of such a model study, recommendations for a changed management can be derived, for example, reduced tillage with simultaneous increased supply of organic matter. Such system and discipline overarching research and state of the art data approaches (AI-research, modelling, standardized big data) are required to provide solutions for a sustainable agricultural production [7, 8]. However, this is only possible on a large scale if data are available and found in a utilizable form which in turn requires an advanced understanding of different data ecosystems with their specific stakeholders and data culture, state of the art research data management (RDM) methods and places high demands on the data infrastructure [9]. The various sub-disciplines in agricultural sciences and the different scales (e.g., from gene over plant and field to landscape) on which agricultural research is conducted, but also the organizational aspects of the research landscape in Germany pose particular challenges here.

In Germany, scientific research in the agricultural domain is quite scattered and mainly carried out by universities and universities of applied sciences as well as by federal institutes, institutes of the Leibniz Associations and other more regional research institutions [10]. There, it covers research areas and disciplines, such as crop, livestock and soil sciences, forestry, agricultural economics and sociology, applied genetics and physiology, research on genetic resources and breeding, plant and animal pathology, horticulture and viticulture. Moreover, it covers systems-oriented research on agroecosystems, ecosystem services and agricultural landscapes and represents a real system science [11]. New and innovative methods of data acquisition and the digitalization of legacy data (e.g., data from long term experiments in the BonaRes Repository [12]) lead to a constantly growing availability and accessibility of data. Emerging trends and developments in the field of Internet of Things (IoTs) and sensor technology, robotics and automation (i.e. smart agriculture, Farming 4.0), artificial intelligence and machine learning are paving the way to more data-driven approaches [8, 13, 14]. Despite important international initiatives addressing data sharing and reusability like GODAN [15] or AIMS of the FAO [16], in Germany such topics have only recently been discussed in agricultural science. Furthermore, the number of respective data sharing infrastructures are small here, differ in standards, are not connected and most importantly not intensively used or known by the wide community of data producers [17]. In order to improve the reusability of scholarly data it was recognized that it needs a concise and measurable set of principles with specific emphasis on enhancing the ability of machines to automatically find and use data, in addition to supporting its reuse by individuals–namely the FAIR principles (Findable, Accessible, Interoperable, and Reusable [18, 19]). They were designed and jointly endorsed by a diverse set of stakeholders representing academia, industry, funding agencies, and scholarly publishers [20, 21].

This has also been recognized by major national and international funding agencies (e.g., U.S. National Science Foundation and the National Institute of Health, and the European Research Council) identifying the enormous value of FAIR research data and are paying more and more attention to making them available for open and free reuse. Therefore, a well-managed RDM with the aim of data publication is increasingly and comprehensively required by international and national funding agencies such as the EU (e.g. Horizon Europe) or DFG (e.g. NFDI), but also individual federal programs such as the BMBF (e.g. National Bioeconomy

Strategy) or BMWi (e.g. GAIA-X) in recent years. An important milestone in bringing RDM into the awareness of our users in the agricultural sciences is the NFDI—a digital, distributed national research data infrastructure [22]. It is currently under development and will provide services and advice to the whole scientific community in Germany on all aspects of RDM and to systematically make publicly funded research data accessible, sustainably and to make it state of the art retrievable [23].

There is no doubt that a scalable infrastructure must be provided for the storage, provision and reuse of research data. In order to be able to build such an infrastructure, the specific requirements of the users must first be evaluated. Especially in the domain of agricultural sciences, there is no common understanding among typical users what RDM following the FAIR principles means [19]. Often, the mutual benefits of FAIR data provision remains unrecognized by both data creators and (re-)users [24, 25]: more and secured storage, time and cost savings, facilitating validation, reproducibility and reuse consequently avoiding duplicative studies and accelerating research, increasing citability and reputation. Additionally, it is increasingly requested by funding agencies and becomes a prerequisite of data papers or scientific articles. State of the art RDM along all stages of the data life cycle should cover data creation, data curation, and data provision, respecting authorship rights and privacy concerns when dealing with sensitive data [26, 27]. A domain-specific RDM can contribute to making accessible the knowledge that is often implicitly contained in the data but not or only sparsely exploited.

The involvement of scientific users is a success factor for the development of an efficient common research data infrastructure [28]. Therefore, the first step towards a sustainable solution must be to a) identify the potential users b) identify the data its format and structure and the way how it is managed: created, stored, documented, published and shared c) query their needs and wishes to develop a tailored common research data infrastructure for agricultural science and suitable RDM. An online survey, which the authors conducted in the German agricultural science community in 2020, serves as the means of choice for this purpose. Our study inquires not only about the status quo, but also explicitly about the wishes and needs of users representing the agricultural scientific research domain. Thus, this survey complements previous surveys on RDM status in Germany with agricultural scientists as participants that were mainly conducted at universities at the individual or state level. However, the latter did not specifically target agricultural scientists (<70 agricultural scientists as participants) and the number of questions on concrete needs has so far remained relatively small (e.g., [29–33]).

## Methods

In order to develop an RDM along the needs of involved user groups, we first defined the relevant user groups in a selection process as follows: *data users*, *data providers* as well as *infrastructure service* and *information service providers*. *Infrastructure service providers* cover for instance operators of data repository and databases as well as of data management and analysis tools. *Information service providers* are typically librarians or data stewards with high data literacy and expertise in training, support and education in RDM. We developed a questionnaire addressing these different user groups in or related to agricultural science in Germany. The questionnaire is available in [34]. It contained a cover page providing the background of the survey. The online survey was voluntary, could be completed anonymously and be discontinued at any time. We collected no ethically relevant information (e.g., about gender, age, nationality, health, etc.). Therefore, IRB approval was not necessary. At the end of the questionnaire, participants could optionally provide their name, institution, e-mail or telephone in case they wanted to be contacted. This required an informed consent in the form of a check

box from the participants containing a GDPR-consistent data protection and privacy disclosure declaration including the right to withdraw their data. In case personal information was given this was not analyzed together with the questionnaire data and are not presented here or the data supporting this study itself. Furthermore, in the data files supporting this study individual answers were extracted from the main data set and presented separately to be fully anonymized [34].

In total, the questionnaire contained 52 questions (Q1-Q52) on information about produced and (re-)used data, data quality aspects, information about the use of standards, publication practices and legal aspects of agricultural research data, the current situation in research data management (RDM) in regards to awareness, consulting and curricula as well as needs of the agricultural community in respect to future developments. Questions were partly user group specific while the first general set addressed the respondent's professional background (i.e. career status, affiliation and subject area, no personal data) and the user group. To conduct the survey we used the Community Edition of the Open Source Survey Tool LimeSurvey (Version 3.19.3; LimeSurvey GmbH, Hamburg, Germany). The survey was accessible online between June 26$^{th}$ and July 21$^{st}$ 2020 and took about 20 minutes. To reach the broadest possible agricultural science community, we promoted the survey in an undirected manner via numerous mail lists of agricultural institutes and agricultural-specific professional societies in Germany, as well as via social media (e.g. Twitter) and announced it at the first community workshop of NFDI4Agri with 126 participants on July 15$^{th}$ 2020 and other related scientific events.

After closing the survey, we exported the data from the LimeSurvey tool (csv, RData and syntax files) and initially screened it. We considered all questionnaires that contained at least one answered question in addition to the respondent's professional background information. The data set is available in [34]. In the result section we give absolute numbers of responses in parentheses where percentages are given in the text as some questions were only for a subset of respondents or some questions were skipped or not all answer options were always used by all possible respondents. If there was the option to enter free text in addition to given answer options, we standardized them if they occurred more than once. Otherwise, we summarized them under the category "other" or partly cite them individually. Analysis and graphs were done using R [35] and the R packages "likert" [36] and wordcloud [37]. We used $X^2$-square tests to compare answers given, e.g. proportion of "yes" and "no" answers and Mann-Whitney-U-Test/ Wilcoxon-Tests (WRS) to compare five-point likert items [38]. However, since this study is more of a status quo analysis of RDM in the agricultural sciences, we largely avoid comparative analysis and statistics (e.g. between user groups, etc.). Graphs were arranged with the vector-based program Inkscape [39].

## Results and discussion

### Participants at a glance

In total we received 196 questionnaires of which 160 were completed in full. 152 of the respondents assigned themselves to the group of *data providers*, 161 to *data users* (128 are *both data providers* and *users*), 20 to *infrastructure service providers* and 13 to *information service providers* (Q1; multiple answers were possible). The majority of respondents are scientists (115), followed by professors or group leaders (64), non-scientific staff (8) and others (8; Q2). Most are affiliated with non-university research institutions (161), followed by members of universities/colleges/universities of applied sciences (21) and industry (3) and others (13; Q3; multiple answers were possible) and cover the plant, animal, soil, water and geological disciplines (Fig 1, Q4). Thirteen *infrastructure service providers* operate internal institutional repositories; four

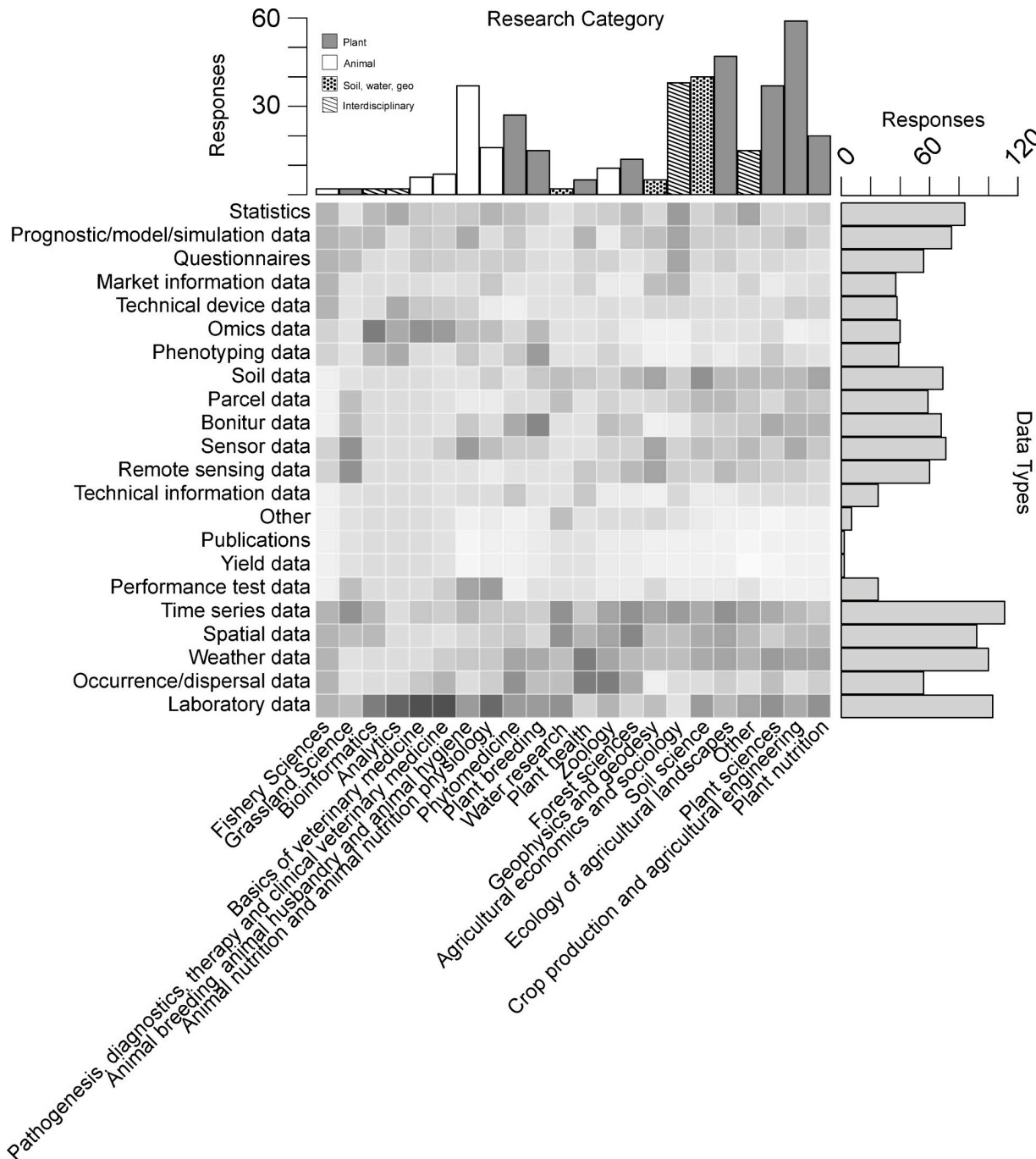

**Fig 1. Data types in different agricultural disciplines.** The horizontal barplot on top showing the number of respondents that assigned themselves to each of the different agricultural disciplines (Q4). The vertical barplot on the right shows the number of respondents working with the different types of data (Q10). The heatmap shows the relative distribution of data types among the different research categories (N = 191). For better comparability of the various research categories, the values were centered and scaled in each column (i.e. research categories).

operate a public institutional repository. Three operate a public discipline-specific repository and one operates a server-based and internal software solution for customers (Q15).

The result that the overwhelming majority of surveyed scientists are affiliated with non-university institutions compared to the very low part from universities reflects not the actual

distribution of scientists among these affiliations which is in reality rather balanced [10]. The reason for this might be that the propagation of the survey was mainly pushed and distributed by the authors using their networks, scientific societies as well as contacts to other non-university research institutions. However, the focus on researchers from non-university institutions will complete the picture of RDM practice in Germany, since most surveys covering *i.a.* agricultural sciences (e.g., [29, 30, 40–44] targeted mainly universities.

### The current RDM situation in agricultural sciences

**The diverse research data landscape.**   92.1% of *data providers*, *users* and *infrastructure service providers* (N = 191, Q11) reported dealing with numerical data, 57.6% with text data, 39.8% with photographical data, 38.2% with geographic data, 27.2% with source codes, 13.6% with video material, 7.3% with audio data. 1.6% stated dealing with other digital data like provenance data. 36.1% work with non-digital data. Across all research disciplines in agriculture respondents often work with data containing time series (58.1%), laboratory (53.9%), weather (52.4%), and spatial content (48.2%; N = 191; Q10; Fig 1). Statistical (44.0%), model (39.3%) and questionnaire data (29.3%) and occurrence/dispersal data (29.3%) also represent a large part of data across disciplines. Especially in the field of plants, respondents frequently work with sensor data (37.2%), soil analysis data (36.1%) bonitur (35.6%), remote sensing (31.4%) and parcel data (30.9%; N = 191; Q10; Fig 1). In the animal sector respondents frequently work with omics (20.9%), phenotype (20.4%) and technical device data (19.9%; N = 191; Q10; Fig 1). It shows how diverse the agricultural sciences are in terms of data types, covered scales and disciplines and gives a taste of the challenges that exist in developing a common RDM across this community.

When asked what data volume is generated, or if representative of a workgroup, what is the average data volume generated by a typical employee per year, 25.4% of *data providers* (N = 130; Q12) reported producing less than one GB per capita per year. 59.2% reported producing between one GB and one TB, 13.1% produce one to ten TB and 2.3% even generate over ten TB p.a. To store their data *data providers* as well as *data users* (N = 185; Q13) use in most cases an institution's own internal storage (93.0%). 66.5% use local storage media. Percentages of respondents storing their data in publicly accessible institutional repositories/databases (16.2%), publicly accessible topic-specific repositories (11.9%) or generic repositories (7.0%) are rather low. The reported data volumes and storage habits are consistent with data from other surveys that have also covered the agricultural sector. Accordingly, 2% of agricultural scientists surveyed as part of the UNEKE survey at German universities collect less than one GB, 73% one GB to one TB, 18% one TB to ten TB and 7% more than ten TB (N = 45; [29, 44]). Among agricultural scientists at the University of Giessen the usage of local storage media like hard disks on business and private PCs or external storages like USB-sticks, CD/DVD is also very common during ongoing research processes and projects ([33]; 16–21 of 22 respondents answered with "do it always" or "do it sometimes"). This complicates a documented versioning, standardized metadata description, safe backup and sharing already during data generation.

**The basic awareness towards RDM in agricultural sciences.**   Through their work, *information service providers* often have a good overview of the RDM situation in numerous disciplines. When asking them (N = 13, with two who could not estimate this; Q27) about the fundamental awareness of researchers, especially with regard to data description, archiving and publication at their institution, only one says, "Most of them already practice RDM according to the FAIR principles". While five *information service providers* say "most people are aware of the necessity, but only a few practice it", another five say "a large part of them

must first be convinced of the necessity and there is a lack of support and advice". This estimation is also reflected in the following numbers around the publishing and reuse of research data answered by *data providers* and *users*.

**Publishing and sharing research data habits.**    While 4.8% of *data providers* (N = 146; Q16) do not make their data publicly available in any way, 66.2% at least sometimes make data available as part of an independent publication (N = 145), 78.8% to support a research paper (N = 146) and 61.1% by providing metadata (i.e. data itself is not published; N = 144; Fig 2A). However, it is far from being the rule: only 13.0–26.2% always publish data in one of the previously mentioned ways (Fig 2A). This coincides with the results of a survey of the University of Giessen in Germany, where only 5 of the 22 surveyed scientists said that making data publicly available is an established practice in their discipline, however, 14 wished that it would be [33]. Reasons for not publishing include privacy reasons for 62.4%, business interests for 48.9%, and research ethics for 14.9% of the *data providers* (N = 141; Q18; Fig 2B). Interestingly, 9.2–23.4% (N = 141) were not even sure whether the aforementioned reasons were the reason for not publishing, reflecting a high level of uncertainty in this area. (Fig 2B). The need for clarification here is also shown by the fact that 91.9% of the *data providers* who cannot make their data freely available (N = 99; Q19) estimated that this data has a high (51.5%; at least partially relevant for other researchers) or at least medium (40.4%; suitable for subsequent use within working group or institution) reuse potential.

In addition to the possibility of sharing data publicly, *data providers* and users exchange such sensitive data in other ways. On request 28.3% of the *data providers* (N = 99; Q20) usually provide such data directly to colleagues. "Unfortunately, however, only very few research colleagues know that there is data that could possibly be relevant for them" and "finding a data

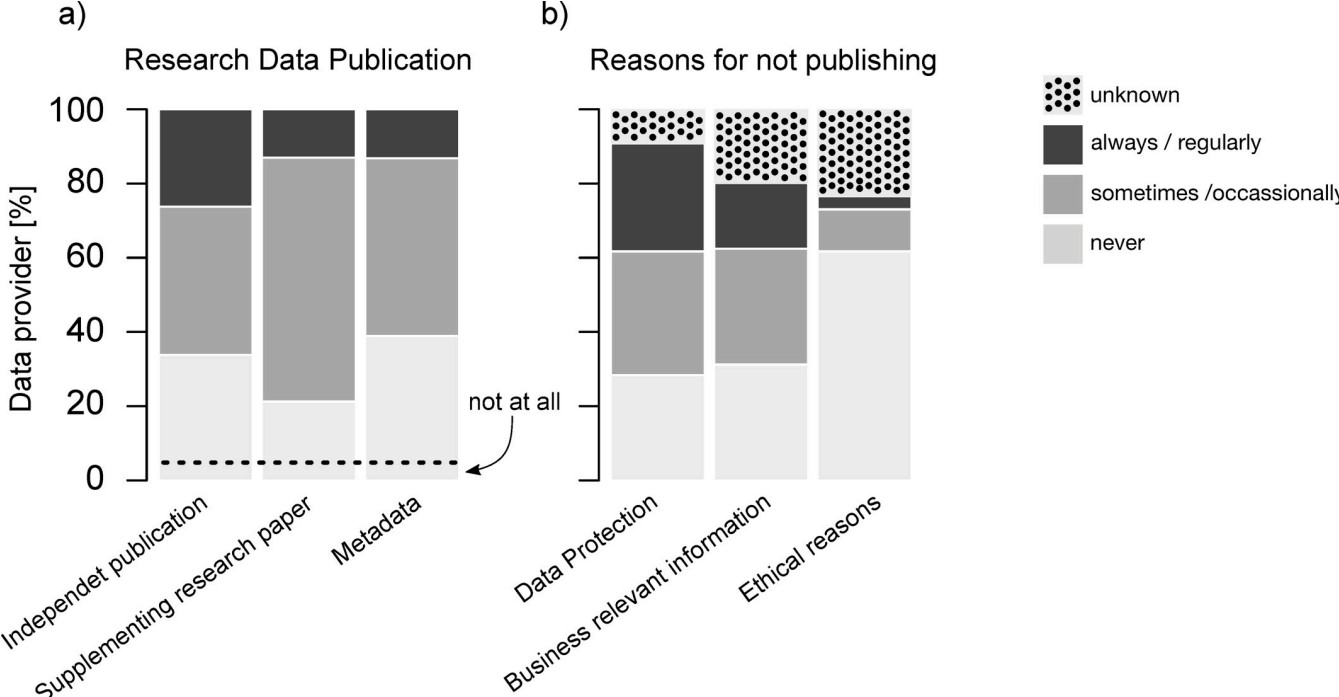

**Fig 2. Research data publications. a)** Percentage of *data providers* publishing their data always, sometimes or never in one of the mentioned ways (independent publication, supplementing a research paper or by providing metadata (i.e. data itself is not published). N = 144; N = 145, N = 143 respectively (Q16) **b)** Reasons for not publishing research among *data providers* that cannot make their data available are data protection, business interests or ethical reasons: regularly, occasionally, never or unknown: N = 141 (Q18).

protection-compliant regulation is usually very time-consuming" (two respondent's comments). 52.5% of *data providers* only partly share their research data, as it is usually only usable and exchanged in the context of a research cooperation. 13.1% do not usually share such data due to legal or ethical concerns and 6.1% do not share any data so far.

**Describing research data.** When actual data sets are provided, 55.0% of *data providers* (N = 109; Q43) also often or even always provide metadata on sources, terms of use, authors, processing methods, etc. Occasionally metadata are provided by 18.3% and 26.6% never provide metadata for their data sets. 7.5% of the *data provider* (N = 80; Q44) providing metadata reported using the Data Cite, 2.5% the OGC/ISO 19115 and 1.3% use the Dublin Core metadata schema. Four *data providers* indicated using their own metadata standards or documentation (including annotations in documents or tables). One person each indicated using MIAME or INSPIRE schemas. However, most (75.0%) are not aware of the metadata standards used. The survey paints a similar picture for data standards that can be used to describe data sets (i.e. ontologies). 84.7–96.8% of *data providers* (N = 92–99; Q45; across all disciplines) do not know the following data standards: Environment Ontology (ENVO), Multi-Crop Passport Descriptors (MCDP), Animal Trait Ontology for Livestock (ATOL), OGC standards, Plant Trait Ontology (TO), Minimum Information (MI) Standard for Plant Phenotyping (MIAPPE), Plant Ontology (PO), ISO11783, Crop Ontology (CO), AGROVOC or Gene Ontology (GO). Most *data providers*, *users* and *infrastructure service providers* (81.4%) do not use or know (16.8%, N = 161; Q47) Linked Data /Semantic Web Technologies. The percentage of those who use it ranges between 1.2–3.1% (in ascending order: SPARQL query endpoints, ReSTful Web Services with RDF formats, OWL ontologies, RDF scheme for data description). Among them are three *infrastructure service providers* (N = 16) using these technologies. Regarding non-semantic technologies, 59.6% of *data providers*, *users* and *infrastructure service providers* do not use and 21.1% do not know them (N = 161; Q48). The percentage of those who use it ranges between 0.6–19.3% (in ascending order: OpenAPI/Swagger, Google protobufs, WSDL/SOAP, JSON and XML-Schemes). Among them are six *infrastructure service providers* (N = 16) using WSDL/SOAP, OpenAPI/Swagger, JSON and XML-Schemes.

27.4% of the *data providers* (N = 95, Q49) use persistent identifiers to identify information resources. Most of them use DOIs (92.3%, N = 26; Q50). Others are less frequently used, like ePIC (3.8%) or own URN Schemes (3.8%) or not used (PURL) or data providers do not know which one they use (7.7%). Further identifiers mentioned by respondents individually were ORCID, ISNI, GRID and ISBN which are PIDs mostly used for entities other than research data.

**Searching for research data in agricultural sciences.** 83.7% of *data users* and *providers* (N = 153; Q37) regularly use generic search portals like Google (e.g. Scholar or DatasetSearch, Google LLC, Mountain View, California, USA) to search for relevant data and literature publications (Fig 3A). 50.0% (N = 152) use social/professional networks like ResearchGate (ResearchGate GmbH, Berlin, Germany). 42.3% (N = 149) use subject-specific search portals, e.g. LIVIO (ZB MED–Information Centre for Life Sciences, Cologne, Germany), BonaRes [12], PubMed (United States National Library of Medicine (NLM) at the National Institutes of Health, Bethesda, Maryland, USA) or Pangaea (Alfred Wegener Institute, Helmholtz Center for Polar and Marine Research (AWI) and Center for Marine Environmental Sciences, University of Bremen (MARUM), Germany). Only 34.9% (N = 149) regularly use commercial databases like Scopus (Elsevier, Amsterdam, Netherlands) or Publons (Clarivate Analytics, London, UK). Clear and sensitive search terms make it easier to find relevant data or articles, especially in an interdisciplinary context. Formulating such search terms is easy for 68.8% of the respondents (N = 160; Q38; Fig 3B).

a)

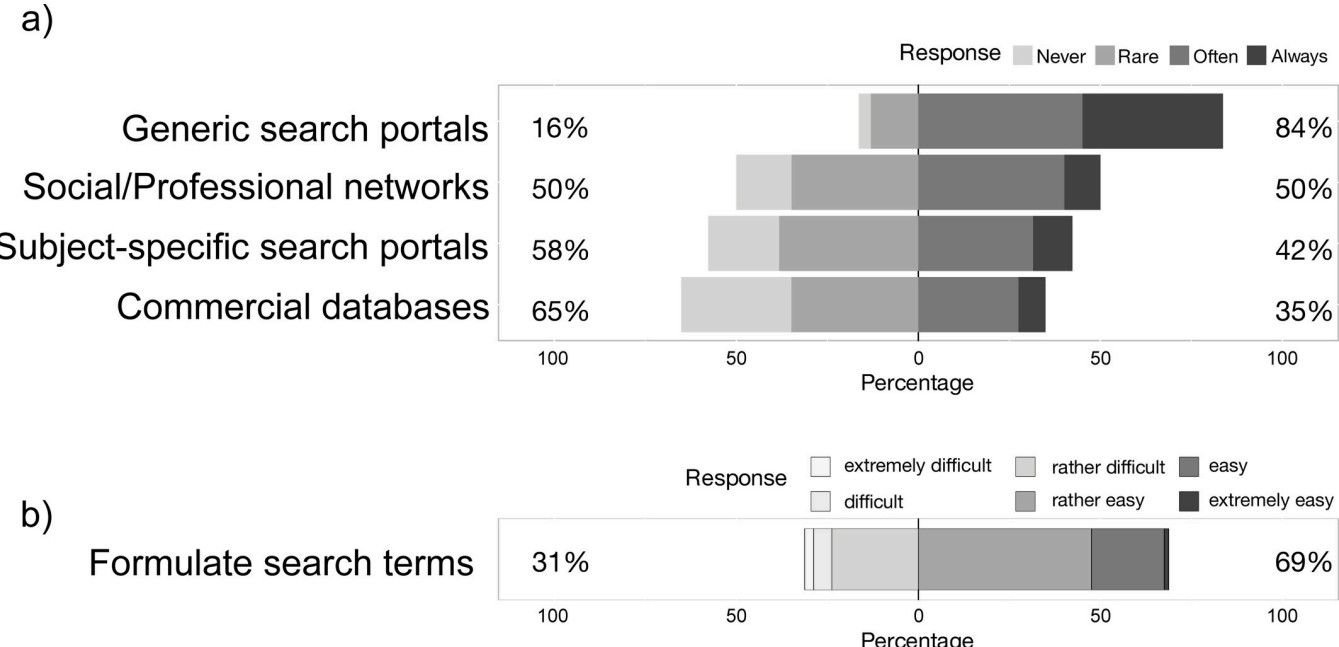

**Fig 3. Searching for research data.** a) How often do *data providers* and *users* use different available search tools to search for relevant data and literature publications (N = 149–153; Q37)? b) Respondents' assessment of how they rate the formulation of clear and sensitive search terms when searching for relevant data and articles, especially in interdisciplinary context (N = 160; Q38).

**Data quality and provenance and its obstacles.** Not all data is equally suitable for use or re-use. Fig 4 shows what matters to *data users* in terms of quality aspects (N = 67–126, Q29; not all *data users* responded to every listed quality aspect). Among the top five rated data quality aspects are "plausibility of content", "secure rights of use/open access", "verified consistency of content", "information on data pretreatment" and "content completeness". Less important seem to be "spatial uncertainties (e.g. Kriging Standard Variance or "Random Forest Probability")", the "up-to-dateness" and "spatial completeness" of the data (Fig 4).

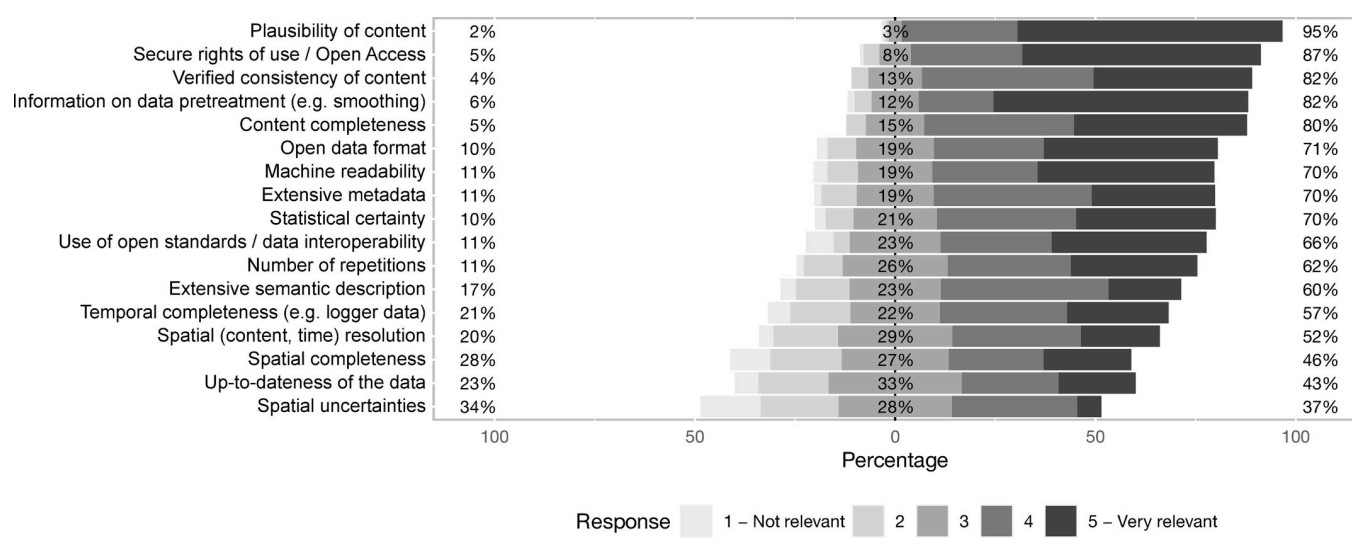

**Fig 4. Important data quality aspects.** Likert plot of data quality aspects that are important to *data users* (N = 67–126; Q29).

58.3% of *data users* also always or at least often analyze metadata, when reusing data from others (e.g. for source analyses, quality checks, etc.; N = 115; Q40). Occasionally, metadata are analyzed by 25.2% and 16.5% do not use metadata for further analyses. Among *data users* (N = 96; Q41) who use metadata at least occasionally (i.e. including always and often), the following metadata were perceived as relevant: "parameters of the data set (e.g. date of survey, sample number, soil temperature, etc.)" (85.4%), "technical description of the data source (e.g. sensor, web service, etc.)" (79.2%), "description of processing steps when generating the data set (e.g. statistical cleaning of outliers, aggregation over time intervals, etc.)" (79.2%), "originator/author of the data (e.g. person X, institute Y)" (66.7%), "information on the data format (e.g. txt, csv, etc.)" (52.1%) and "the scope of the data set (X TByte)" (24.0%). Other statements made by individual respondents included "site parameters", "taxonomic background", "calibration routines", and "implicit hypotheses".

**Advice and support services in agricultural sciences.** 50.0% of *information service providers* (N = 12; Q5) reported providing agriculture-specific advice on RDM, while 50.0% only provide general advice on this topic. Legal aspects are only generally advised by 30.0% (N = 10). Regarding infrastructures and software, 45.5% gave agricultural-specific advice (N = 11), 36.4% general advice and 18.2% no advice at all. When asked about the topics addressed by agricultural scientists 61.5% of the *information service providers* (N = 13; Q6) said "publications", 53.8% "data organization", 53.8% "metadata", 46.2% "repositories", 38.5% "IT related questions", 38.5% "data standards", 30.8% "archiving", 30.8% "DMPs", 23.1% "legal issues/privacy", 23.1% "analysis" and 7.7% "licenses". None of the *information service providers* (N = 13; Q25) indicated that RDM is an integral part of agricultural science (and related disciplines) curricula at their institution so far or that this is planned.

## RDM needs and wishes of the agricultural community

In the following, we address ways to improve the situation in agricultural sciences with respect to a common RDM and to make data available in a FAIR, quality-assured, and sustainable way. To this end, we asked the study participants for explicit technical services and support that should be developed or offered from the community's point of view.

**Enhance data publications.** To increase the number of data publications, user-friendly user interfaces, sovereignty and security aspects as well as the necessary incentives are the most helpful factors. 64.4% of *data providers* (N = 146; Q17) are in favor of a "data upload via a graphical user interface" (64.4% responded with "yes"; $X^2$ = 12.082, df = 1, p<0.001) as technical component that would help them to publish research data (Table 1).

"The required provision of data should thereby be integrated into the daily workflow seamlessly and without additional effort", e.g. through "automatization" (two individual answers; Q52). In doing so, they would trust their own institution the most. Thus, *data providers* would be more likely to rely on either a customizable software solution (median of 4 on a scale from "1 –not at all" to "5 –Very much"; N = 126; Q14) or individual software components expanding the existing data infrastructures (median of 4 on a scale from "1 –not at all" to "5 –Very much"; N = 122; Q14; WRS between both options: p = 0.407) to be installed and hosted by their institution. In both cases we assumed that their data would be stored on the servers of their institution and can be provided with access rights; the basic functions are supported by the community. In comparison, an externally managed and developed portal (cloud, e.g. OpenScience Framework) where their data would be stored externally, one can assign access rights and individual adjustments would not be possible, was less preferred (median of 3 on a scale from "1 –not at all" to "5 –Very much"; N = 128; Q14; WRS with either of the preferred options: p<0.001). Last but not least, "data publications should be taken into account in

**Table 1. Technical components supporting publication of research data.**

| Question 17: The following technical components would support me in publishing my research data: | Yes [%] | $X^2$ | df | p |
|---|---|---|---|---|
| Data upload via a graphical user interface | 64.4 | 12.082 | 1 | <0.001 |
| Metadata description via a guided graphical interface | 55.5 | 1.753 | 1 | 0.185 |
| Automatic quality checks of data (e.g. outlier test, plausibility check, etc.) and metadata | 54.1 | 0.986 | 1 | 0.321 |
| Service for online access to own data sources. | 44.5 | 1.753 | 1 | 0.185 |
| Web services for API access (e.g. batch-processed mass upload of data, metadata) | 37.0 | 9.890 | 1 | 0.002 |
| Metadata description via uploaded metadata files (e.g. XML files) | 35.6 | 12.082 | 1 | <0.001 |

Answers given by *data providers (N = 146) at question Q17*. $X^2$-square tests were used to compare answers given, i.e. proportion of "yes" and "no" answers. Answer options with significantly more supporters ("yes") are displayed in dark gray, with balanced answers ("yes" / "no") are displayed in medium gray and answer options significantly more rejected ("no") are displayed in light gray.

performance evaluation (institute evaluations, evaluation of officials, etc.)" (65.1% of *data providers* and *users*, as well as *information service providers* responded with "yes"; N = 175; Q28; $X^2 = 16.051$, df = 1, p<0.001). In contrast, survey participants reject a certain urge to publish or financial benefits for data publications (Table 2). Further individual measures mentioned are providing user-friendly publication services (upload and findability), appropriate citability and publication performance indicators for data, seamless data collection workflows as well as additional supporting staff (Q28).

**Facilitate data re-use.** In order to fuel the re-use of data, a definition or common "specification of metadata standards" (N = 1, Q52), "well-structured and extensive search options" (N = 2, Q52), e.g. "targeted searches for spatial data" (N = 8; individual answers; Q39), "climate data", "species occurrence data", "image data", "excel and pdf files" (individual answers; Q39), "simplified access from analysis software, e.g. R (N = 1, Q52)" but also "simplified access to previously inaccessible datasets" (N = 1, Q52) were individual suggestions. Among technical tools that would help to formulate search queries, "keyword search via interactive graphical user interface with suggestion system and navigation" was favored by *data* (re-)*users* (79.9% responded with "yes"; $X^2 = 49.561$, df = 1, p<0.001, N = 139; Q39) formulating search queries.

**Table 2. Incentives to encourage researchers to publish data more frequently at their institution.**

| Question 28: What incentives would you like to see established/developed to encourage researchers to publish their own data more frequently at their institution? | Yes [%] | $X^2$ | df | p |
|---|---|---|---|---|
| Data publications should be taken into account in performance evaluation (institute evaluations, evaluation of officials, etc.). | 65.1 | 16.051 | 1 | <0.001 |
| Data providers should be offered co-authorship, if data is subsequently used. | 57.1 | 3.571 | 1 | 0.0588 |
| Appropriate metrics reflecting the quality/post-use of the data. | 53.7 | 0.966 | 1 | 0.326 |
| Publication of data parallel to the article should be obligatory according to the institution's data policy. | 39.4 | 7.823 | 1 | 0.005 |
| Publishers/funders should make data publication mandatory. | 34.3 | 17.286 | 1 | <0.001 |
| Publication of data should be remunerated on a monetary basis. | 13.1 | 95.091 | 1 | <0.001 |

Answers given by *data providers* and *users*, as well as *information service providers (N = 175) at question Q28*. $X^2$-square tests were used to compare answers given, i.e. proportion of "yes" and "no" answers. Answer options with significantly more supporters ("yes") are displayed in dark gray, with balanced answers ("yes" / "no") are displayed in medium gray and answer options significantly more rejected ("no") are displayed in light gray.

The "support of query languages" (73.4% responded with "no"; $X^2$ = 30.396, df = 1, p<0.001) and an "application programming interface for use in own software" (81.3% responded with "no"; $X^2$ = 54.453, df = 1, p<0.001) were considered as less helpful. This was probably because the direct benefits were not clear to the respondents, as these were very technical statements.

Further technical components that could help to foster data re-use, are options for "comparisons of objects according to multiple properties" as well as "sorting and prioritization on a broad data basis (e.g. variety comparisons)" (individual answers; Q52).

**Ensure data quality.**   "Data validation", such as "plausibility checks" can be used to foster data re-use (two individual answers; Q52). 80.8% of *data providers*, *users* and *infrastructure service providers* (N = 130; Q30) would endorse tools for validating data, for instance to identify crude errors ($X^2$ = 49.231, df = 1, p<0.001). The word cloud presented in Fig 5 shows data for which the respondents could imagine using such a validation tool (N = 66 and 104 mentioned data types, 47 different data types; Q31).

Furthermore, *data providers* and *users* positively rated a potential infrastructure service that links raw data and processed data with a processing description (median of 4 on a scale from "1 –not at all" to "5 –Very much"; N = 128; Q32).

**Secure data re-use.**   Often, data exchange does not occur or data is not made available to the public due to legal concerns or uncertainties. In addition to "clear data sharing policies" and "information on data protection aspects", tools, for instance "a tool for legally secure anonymization of georeferenced data" (individual answers, Q52) could help to make data provision and re-use more secure. In economic and social sciences, it is common practice that research data that cannot be made freely accessible is archived by a data center and, under certain conditions, made available exclusively for scientific re-use [45]. These data centers also provide advice on legal methodological and technical aspects of handling confidential research data (e.g. application of data protection, legally binding declarations of consent, effective pseudonymization and anonymization). According to our survey, 12.3% of *data providers* and *users* would use such a consulting service regularly, 72.1% occasionally, and 15.6% not at all (N = 154; Q21). 21.9% of *data providers* would have data archived and released for legally compliant subsequent use in such a trust center on a regular basis, 62.3% at least occasionally and only 15.8% not at all (N = 114; Q23). 13.8% of *data users* would obtain data from such a trust center on a regular basis, 73.2% at least occasionally and only 13.0% not at all (N = 123; Q22). Furthermore, 50% of the *infrastructure service providers* would be interested in building such a system for their database for a significant proportion of their data, 35.7% for a small part of their data and 14.3% do not see any added value in establishing such a service (N = 14; Q24). If there were a demand to reuse sensitive data that was previously stored in a data repository, 50.4% of *data providers* (N = 131; Q42) would support automatic requests to the data provider. 38.9% would prefer an automatic approval system (i.e. data providers do not have to answer every request regarding the provision of their protected data) and for 10.7% it does not matter.

**Train and support.**   When we asked all participants what they would like to see in services to be developed, some of the things that came up were training opportunities, consulting services on statistical methods and data analysis as well as support in curating and preparing research data, e.g. for publication (Q52, individual responses). These statements are supported by a survey on scientific data management among scientists at universities in Baden-Württemberg, Germany [41, 42]. The results of this survey were mapped into 2,554 user stories. These included 219 user stories from agriculture, forestry, horticulture, and veterinary medicine. Among them 43 user stories that expressed a need for more general support and 20 expressed a need for more expert support. 36 would like to see more information and consulting services [42]. 69.2% of the *information infrastructure providers* (N = 13; Q8) stated that external training offers for scientists would help. 46.2% would find training concepts for teachers (i.e. train-

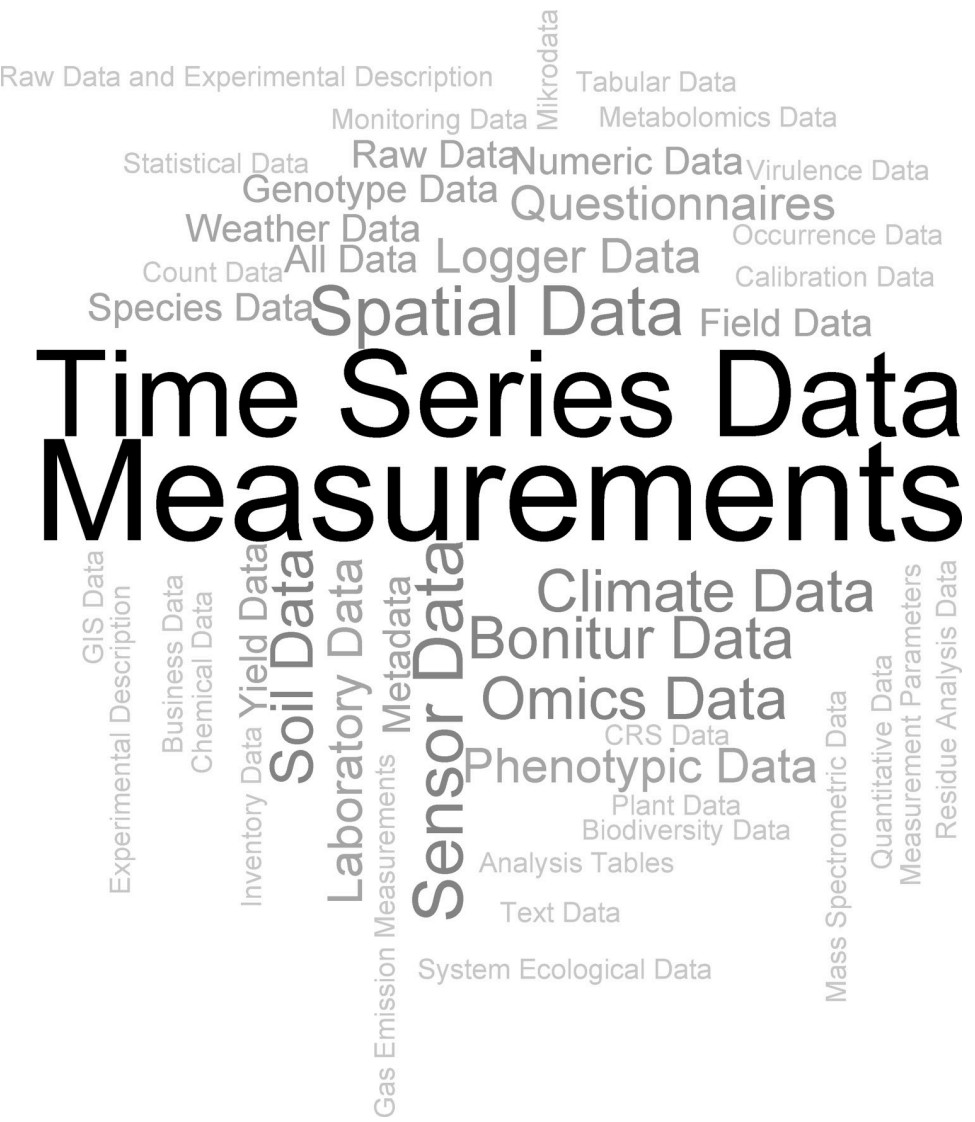

**Fig 5. Wordcloud with data for which the respondents could imagine using a validation tool.** The size represents the frequency with which individual data types were mentioned, e.g. "measurements" was mentioned twelve times (respondents N = 66, 104 mentioned data types, 47 different data types; Q31).

the-trainer concepts) and 38.5% information materials helpful. Other suggestions from *information infrastructure providers* included "networking opportunities", "recognition of the time required for good and sustainable documentation", "development of an infrastructure for data management and archiving", and "general training" and implementation of "best practice projects". To get FDM into the curricula of universities and institutions, *information service providers* suggest providing "subject-specific information materials" and "concepts for RDM curricula", intensifying "networking" and "collaboration with teachers" and "continuing education institutions" (e.g.: graduate centers) (individual responses; N = 6; Q26).

**Bundle services.** There are already a number of offers, services (e.g., infrastructures) and software solutions developed by individual institutions in agricultural sciences (e.g., GBIS/I [46], e!DAL-PGP [47], FarmPheno [48], TSN [49], TERENO [50], plaBiPD [51], OpenAgrar [52], DataCube [53], EmiMin/EmiDat [54], Edaphobase [55], FLOPOknb [56], National Soil

[57] & Forest Inventories [58], PhenoRob Database, BonaRes Repository [12], PUBLISSO Repository for Life Sciences [59], SRADI [60]). However, these are often scattered and do not always find their way to the end user. Therefore, the statement of one respondent to consolidate the previous infrastructures does not come as a surprise (Q52). This respondent proposes to create some kind of portal in which data can be published in different data centers and metadata information is made available centrally, including a search function, links and information on events and training courses. In accordance with this, most respondents regarded an overview of all domain-specific services and associated metrics on quality, availability, and data rather positively. (median of 4 on a scale from "1 –not at all" to "5 –Very much", N = 166; Q51). An individual suggestion included the possibility of user exchange for consultation and exchange regarding data sources and data (N = 1; Q52). 53.8% of the *information service providers* surveyed (N = 13; Q9) believe that a central point of contact (i.e. helpdesk, hotline) to get advice on RDM for themselves or scientists would be useful for agricultural sciences. This is also supported by the statements of *information service providers*, who usually have a broad overview of the RDM situation in the various disciplines, about the current needs in agricultural sciences regarding RDM (individual answers; N = 7; Q7): "harmonization of structures (e.g. metadata, storage and archiving, intellectual property rights); "linking of existing datasets (e.g. time series as well as for long-term monitoring data", "collection and provision of data—also historical data"; "informing scientists more broadly" and demand for "realistic salaries for non-scientific programmers on projects". No matter in which direction the RDM of agricultural sciences will develop in the future, a "good feedback process is desirable"; it is most likely to guarantee that the service is and remains user-friendly (individual statement; Q52).

## Conclusion

### The need for action

One of the most challenging tasks of RDM in agricultural science and especially of a successful integrative data analysis are the many different data types produced and used by the different subdisciplines (Fig 1) and their public availability. A common RDM understanding, standards and interfaces must be developed in the agricultural sciences to overcome these obstacles. In a first step, awareness must be raised regarding the need for good RDM and ultimately the availability and use of standardized data sets. Agriculture-specific education, support for and outreach to *data provider* and *(re)user* in agricultural sciences is needed to fulfil this task. Here, our survey provides starting points for developing and improving such services. It reveals where considerable deficits and uncertainties still exist and that structured and FAIR data handling, descriptions with standardized metadata or the use of controlled vocabularies are far from being the rule. Only a small proportion of *data providers* make their data available, which is probably due on the one hand to a lack of conviction and support, and on the other hand to uncertainties regarding publicly providing data. As far as data (re)user are concerned, the 2014 survey among German universities and research institutes (including 58 agricultural scientists) showed that, compared to others, agricultural scientists know far less about where they can find data for reuse [30]. As also shown in our study, most agricultural scientists search generic search portals rather than domain-specific portals, which could significantly limit search success. Topic-specific data repositories and infrastructures, like German research in general in the agricultural sciences, are often scattered, heterogeneous and incompatible with each other. They do not provide standardized interfaces for information and data exchange. There are already possible solutions for this, such as generic initiatives like schema.org or domain-specific adaptations like bioschemas.org. These examples show a concrete possible direction, but are rarely implemented in practice. Consequently, this is also reflected in the number of data

reusers. According to the study, agricultural scientists have the lowest rate of data reusers compared to scientists from other disciplines (58%, [30]). To improve the situation for agricultural scientists, distributed services must be brought together by common standards and bundled access-wise. The demands on such an infrastructure are high. Potential technical developments should consider user-friendliness and seamless integration in scientific workflows, trustworthiness, ensuring data validation (e.g. plausibility checks, Fig 4), legally sound data reuse and licensing as well as guaranteed data traceability (e.g. by linking processed data with a processing description). Once the various tools are available, findable, usable and finally open to the scientific community, the first step would be to familiarize scientists with the tools and motivate them to use tailored tools. Past experience has shown that this is better done in small steps [61, 62]. As perfectly said by Lowndes et al., 'engagement may best be approached as an evolution rather than as a revolution that may never come' [61]. Therefore, the aim should be to establish a minimum set of tools and techniques first rather than directly aiming to establish perfect RDM practices at once [61, 62]. Last but not least, incentives must be created and communicated. The increased workload that a professional RDM requires, in addition to scientific publishing and acquiring third-party funding, must be acknowledged as output in the agricultural science community and the respective institutions, i.e. expanding the concept of reputation by including data publications or development of infrastructures managing data [63]. All of this can only be done through a cultural change, which we are just at the beginning of. National and international funding agencies have recognized the necessity and are increasingly making the publication of data an integral part of the evaluation of projects.

Although this survey was conducted on a national level in Germany, the results of this study can help to address overarching challenges in RDM and professionalize it on an international level as well. RDM is not a national concern, because only together the common challenges in RDM can be solved and will advance global research in agricultural sciences making it at the same time more transparent and efficient.

## Acknowledgments

We thank the whole NFDI4Agri / FAIRagro Consortium (www.nfdi4agri.de), especially Daniel Arend (IPK, Gatersleben), Franziska Boehm (FIZ, Karlsruhe), Heinrich Cuypers (FBN, Dummerstorf), Juliane Fluck (ZB MED, University of Bonn, Bonn), Frank Ewert (ZALF, Müncheberg), Christoph Germeier (JKI, Quedlinburg), Niklas Hartmann (University Potsdam, Potsdam), Thomas Hartmann (FIZ, Karlsruhe), Jan-Henrik Haunert (University Bonn, Bonn), Florian Hoedt (Thünen, Braunschweig), Matthias Lange (IPK, Gatersleben), Birte Lindstädt (ZB MED, Köln), Daniel Martini (KTBL, Darmstadt), Markus Möller (JKI, Braunschweig), Jochen Christoph Reif (IPK, Gatersleben), Xenia Specka (ZALF, Müncheberg) and Claus Weiland (SGN, Frankfurt) for providing questions and comments during the survey design and for critically reviewing the manuscript.

## Data accessibility

The questionnaire and data set with the survey results are published under a CC-BY license in OpenAgrar [34].

## Author Contributions

**Conceptualization:** Matthias Senft, Ulrike Stahl, Nikolai Svoboda.

**Data curation:** Matthias Senft.

**Formal analysis:** Matthias Senft.

**Investigation:** Matthias Senft.

**Methodology:** Matthias Senft, Ulrike Stahl.

**Project administration:** Matthias Senft.

**Validation:** Matthias Senft.

**Visualization:** Matthias Senft.

**Writing – original draft:** Matthias Senft, Ulrike Stahl, Nikolai Svoboda.

**Writing – review & editing:** Matthias Senft, Ulrike Stahl, Nikolai Svoboda.

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
