## [Decision Letter · Decision Letter 0]

20 Jul 2022

PONE-D-21-39572Research Data Management in Agricultural Sciences: We are not yet where we want to be.PLOS ONE

Dear Dr. Senft,

Thank you for submitting your manuscript to PLOS ONE. After careful consideration, we feel that it has merit but does not fully meet PLOS ONE’s publication criteria as it currently stands. Therefore, we invite you to submit a revised version of the manuscript that addresses the points raised during the review process.

We look forward to receiving your revised manuscript.

Kind regards,

Cataldo Pulvento

Academic Editor

PLOS ONE

Journal Requirements:

https://journals.plos.org/plosone/s/file?id=ba62/PLOSOne_formatting_sample_title_authors_affiliation
**
s.pdf
**

Additional Editor Comments (if provided):

Dear Authors

the paper will be be evaluated for publications on PLOS ONE after minor changes according to the reviewers comments.

Reviewers' comments:

Reviewer's Responses to Questions

**Comments to the Author**

1. Is the manuscript technically sound, and do the data support the conclusions?

Reviewer #1: Yes

Reviewer #2: Yes

2. Has the statistical analysis been performed appropriately and rigorously? 

Reviewer #1: Yes

Reviewer #2: Yes

3. Have the authors made all data underlying the findings in their manuscript fully available?

Reviewer #1: Yes

Reviewer #2: Yes

4. Is the manuscript presented in an intelligible fashion and written in standard English?

Reviewer #1: Yes

Reviewer #2: Yes

5. Review Comments to the Author

Reviewer #1: This manuscript describes and interprets the results of a survey, conducted in 2020, regarding publication, description, and reuse of data in the agricultural sciences. I think that the paper describes important challenges in data management. It presents and interprets the results of the survey straightforwardly. There is not a lot of complexity in the paper to comment on. The writing is somewhat challenging to parse in some places. Editing for clarity and concision would help. I list a few examples here:

Lines 102-103: Something is missing or extraneous in this sentence: "In order to be able to build these infrastructures along the specific requirements of the users must first be evaluated." Perhaps it should be "In order to be able to build these infrastructures, the specific requirements of the users must first be evaluated."

Lines 103-108: This is a long sentence that terminates in a list. The list elements are non-parallel. It would be helpful to break this into two sentences. Also, I am not sure that "data publishing is good scientific practice" is an independent reason. I mean: following FAIR principles is good to do because it is good scientific practice ... is tautological.

Line 127: "In order to develop an RDM along the needs," ... needs of what?

Lines 471-473: This sentence would be more easily parsed if the verb were closer to the subject:

"In a first step, awareness must be raised regarding the need for good RDM and ultimately the availability and use of standardized data sets."

Reviewer #2: # Initial review (2022-07-13)

## Summary

The reviewed study entitled *Research Data Management in Agricultural Sciences: We are not yet where we want to be.* assesses the broad state of data management practices in the crop science community with a survey-based approach. The authors highlight obstacles and challenges limiting data circulation and propose ways forward for building better infrastructure and services in the agricultural sciences field.

## Review

The paper is solid, clearly written and focused on a precise question on a key matter not often addressed within the crop science community. The survey and its analysis enabled the authors to draw a clear picture of the concerned data type, and data sharing and discovery practices. The authors conclude that education on better data management practices is essential in raising awareness and advocating for better technical innovations (standardization, distribution).

The results support the claims of the authors and the discussion is careful.

### General comments

At this step, I don't have precise questions that should be addressed as a prerequisite to the publication of this work. However, I propose a few comments that might add a bit of nuance to the message in this paper.

The first comment is about the validity of the analysis outside Germany. I understand that all the answers to the online survey came from various academic or extension or industry structures in Germany. Maybe the authors could add the country name in the title, and add a sentence that could help readers to generalise the results at least to Europe?

My second comment is about the discussion, more precisely about the way forward to improve research data management. I completely agree with the need to improve the value of experimental data by promoting its reuse. The authors mention education and technical solutions as key levers. However, this section could be enriched with a few insights from literature, to illustrate that acting at a small scale, without aiming for a perfect set of RDM practices might initiate change.

* For example, in *Good enough practices in scientific computing* (https://doi.org/10.1371/journal.pcbi.1005510), Wilson et al. explain that a minimum set of tools and techniques ("good enough" practices) is a first step before trying to set up standards and platforms.

* in *Our path to better science in less time using open data science tools* (https://doi.org/10.1038/s41559-017-0160) Lowndes et al. how open data science tools help to drive a better science, starting at the lab scale.

### Suggestions

In this section, I list points relative to writing style. These points are only suggestions, the authors do not have to respond to these.

> l51. However, state-of-the-art research data management and a clever combination of various useful data offer the potential to move from a GHG source to a carbon sink.

In the introduction section, I suggest toning down this bold assertion, I think that such broad transitions in agricultural systems are not only a problem of research data management, and necessitate multiple points of view on knowledge production. For example, in *A tool for reflecting on research stances to support sustainability transitions* (https://doi.org/10.1038/s41893-019-0440-x) Hazard et al. illustrate how the positivist, reductionist and instrumental stance has proved their ability to solve well-defined problems but that more complex issues, such as sustainability transitions require a scientific pluralism.

6. PLOS authors have the option to publish the peer review history of their article (what does this mean?). If published, this will include your full peer review and any attached files.

Reviewer #1: **Yes: **Steven Cannon

Reviewer #2: **Yes: **Pierre Casadebaig

---

## [Author Response · Author response to Decision Letter 0]

28 Jul 2022

- We also uploaded a pdf with our responses in the upload section -

Reviewer's Comments and Suggestions

Reviewer #1

This manuscript describes and interprets the results of a survey, conducted in 2020, regarding publication, description, and reuse of data in the agricultural sciences. I think that the paper describes important challenges in data management. It presents and interprets the results of the survey straightforwardly. There is not a lot of complexity in the paper to comment on.

The writing is somewhat challenging to parse in some places. Editing for clarity and concision would help. I list a few examples here:

Lines 102-103: Something is missing or extraneous in this sentence: "In order to be able to build these infrastructures along the specific requirements of the users must first be evaluated." Perhaps it should be "In order to be able to build these infrastructures, the specific requirements of the users must first be evaluated."

 L109-110: We rewrote the sentence: „In order to be able to build such an infrastructure, the specific requirements of the users must first be evaluated.“

Lines 103-108: This is a long sentence that terminates in a list. The list elements are non-parallel. It would be helpful to break this into two sentences. Also, I am not sure that "data publishing is good scientific practice" is an independent reason. I mean: following FAIR principles is good to do because it is good scientific practice ... is tautological.

 L110-116: As suggested we broke the sentence into two and changed the listing: „Especially in the domain of agricultural sciences, there is no common understanding among typical users what RDM following the FAIR principles means [19]. Often, the mutual benefits of FAIR data provision remains unrecognized by both data creators and (re-)users [24;25]: more and secured storage, time and cost savings, facilitating validation, reproducibility and reuse consequently avoiding duplicative studies and accelerating research, increasing citability and reputation. Additionally, it is increasingly requested by funding agencies and becomes a prerequisite of data papers or scientific articles.“

Line 127: "In order to develop an RDM along the needs," ... needs of what?

 L 136: We added “of involved user groups“ to that sentence.

Lines 471-473: This sentence would be more easily parsed if the verb were closer to the subject:

"In a first step, awareness must be raised regarding the need for good RDM and ultimately the availability and use of standardized data sets."

 L505-507: We changed the sentence as suggested.

Reviewer #2:

Summary: The reviewed study entitled *Research Data Management in Agricultural Sciences: We are not yet where we want to be.* assesses the broad state of data management practices in the crop science community with a survey-based approach. The authors highlight obstacles and challenges limiting data circulation and propose ways forward for building better infrastructure and services in the agricultural sciences field.

Review: The paper is solid, clearly written and focused on a precise question on a key matter not often addressed within the crop science community. The survey and its analysis enabled the authors to draw a clear picture of the concerned data type, and data sharing and discovery practices. The authors conclude that education on better data management practices is essential in raising awareness and advocating for better technical innovations (standardization, distribution). The results support the claims of the authors and the discussion is careful.

General comments:

At this step, I don't have precise questions that should be addressed as a prerequisite to the publication of this work. However, I propose a few comments that might add a bit of nuance to the message in this paper.

The first comment is about the validity of the analysis outside Germany. I understand that all the answers to the online survey came from various academic or extension or industry structures in Germany. Maybe the authors could add the country name in the title, and add a sentence that could help readers to generalize the results at least to Europe?

 Lines 4-5: As suggested, we added „in Germany“ to the title.

 Lines 543-547: We added the following sentences to the end of the conclusion: „Although this survey was conducted on a national level in Germany, the results of this study can help to address overarching challenges in RDM and professionalize it on an international level as well. RDM is not a national concern, because only together the common challenges in RDM can be solved and will advance global research in agricultural sciences making it at the same time more transparent and efficient.“

My second comment is about the discussion, more precisely about the way forward to improve research data management. I completely agree with the need to improve the value of experimental data by promoting its reuse. The authors mention education and technical solutions as key levers. However, this section could be enriched with a few insights from literature, to illustrate that acting at a small scale, without aiming for a perfect set of RDM practices might initiate change.

• For example, in *Good enough practices in scientific computing* (https://doi.org/10.1371/journal.pcbi.1005510), Wilson et al. explain that a minimum set of tools and techniques ("good enough" practices) is a first step before trying to set up standards and platforms.

• * in *Our path to better science in less time using open data science tools* (https://doi.org/10.1038/s41559-017-0160) Lowndes et al. how open data science tools help to drive a better science, starting at the lab scale.

 Lines 530-536: We added a few sentences including the two suggested references to the manuscript: „Once the various tools are available, findable, usable and finally open to the scientific community, the first step would be to familiarize scientists with the tools and motivate them to use tailored tools. Past experience has shown that this is better done in small steps [61,62]. As perfectly said by Lowndes et al., 'engagement may best be approached as an evolution rather than as a revolution that may never come' [61]. Therefore, the aim should be to establish a minimum set of tools and techniques first rather than directly aiming to establish perfect RDM practices at once [61,62].“

Suggestions:

l51. However, state-of-the-art research data management and a clever combination of various useful data offer the potential to move from a GHG source to a carbon sink.

In the introduction section, I suggest toning down this bold assertion, I think that such broad transitions in agricultural systems are not only a problem of research data management, and necessitate multiple points of view on knowledge production. For example, in *A tool for reflecting on research stances to support sustainability transitions* (https://doi.org/10.1038/s41893-019-0440-x) Hazard et al. illustrate how the positivist, reductionist and instrumental stance has proved their ability to solve well-defined problems but that more complex issues, such as sustainability transitions require a scientific pluralism.

 Lines 57-59: As suggested, we toned that assertion down a bit: „However, state of the art research data management and clever combination of various useful data can, among other things, help to move from a GHG source to a carbon sink.“

 

Journal Requirements

 We adapted respective parts of the manuscript to meet PLOS ONE's style requirements.

 We checked the reference list and updated URL availability and made smaller changes to meet the reference guidelines.

 We corrected reference [25] as we previously referenced to the F1000Reseach review instead to the actual F1000Reseach paper.

 We updated reference [40] as it was no longer available at the previously provided URL.

 We updated reference [53] as the referenced infrastructure was renamed („DaCuRa“ -> „DataCube“) and also changed its URL, we updated that also in the text (L477).

 We added two references suggested by Reviewer 2:

[61] Lowndes JSS, Best BD, Scarborough C, Afflerbach JC, Frazier MR, O’Hara CC, Jiang N, Halpern BS. Our path to better science in less time using open data science tools. Nat Ecol Evol. 2017; 1, 0160. doi: 10.1038/s41559-017-0160.

[62] Wilson G, Bryan J, Cranston K, Kitzes J, Nederbragt L, Teal TK. Good enough practices in scientific computing. PLoS computational biology. 2017; 13(6), e1005510. doi: 10.1371/journal.pcbi.1005510.

 We added one more references (L539-540) as it was published only recently:

[63] Deutsche Forschungsgemeinschaft | AG Publikationswesen. Wissenschaftliches Publizieren als Grundlage und Gestaltungsfeld der Wissenschaftsbewertung. 2022. doi:10.5281/zenodo.6538163.

 We uploaded all figures to PACE and converted them from eps to tif format. There seemed to be everything fine with them. We uploaded the PACE corrected version together with the revision.

 

Explanation for Additional Changes

Changed contact information of the corresponding author

 The corresponding author changed employers in April 2022. We added the current address on the title page to the corresponding author.

---

## [Editor Report · Decision Letter 1]

2 Sep 2022

Research Data Management in Agricultural Sciences in Germany: We are not yet where we want to be

PONE-D-21-39572R1

Dear Dr. Senft,

We’re pleased to inform you that your manuscript has been judged scientifically suitable for publication and will be formally accepted for publication once it meets all outstanding technical requirements.

Kind regards,

Cataldo Pulvento

Academic Editor

PLOS ONE
---

## [Editor Report · Acceptance letter]

20 Sep 2022

PONE-D-21-39572R1 

Research Data Management in Agricultural Sciences in Germany: We are not yet where we want to be 

Dear Dr. Senft:

I'm pleased to inform you that your manuscript has been deemed suitable for publication in PLOS ONE. Congratulations! Your manuscript is now with our production department. 

Kind regards, 

on behalf of

Dr. Cataldo Pulvento 

Academic Editor

PLOS ONE